# The effects of pre-intervention mindset induction on a brief intervention to increase risk perception and reduce alcohol use among university students: A pilot randomized controlled trial

**Natascha Büchele[1], Lucas Keller[1], Anja C. Zeller[1], Freya Schrietter[1], Julia Treiber[1], Peter M. Gollwitzer[1,2,3], Michael Odenwald[1] ***

**1** Department of Psychology, University of Konstanz, Konstanz, Germany, **2** Department of Psychology, New York University, New York, New York, United States of America, **3** Institute of Psychology, Leuphana University Lüneburg, Lüneburg, Germany

* michael.odenwald@uni-konstanz.de

## Abstract

### Objective

Brief interventions based on personalized feedback have shown promising results in reducing risky alcohol use among university students. We investigated the effects of activating deliberative (predecisional) or implemental (postdecisional) mindsets on the effectiveness of a standardized brief intervention, the ASSIST-linked Brief Intervention. This intervention comprises a personalized feedback and a decisional balance exercise. We hypothesized that participants in a deliberative mindset should show better outcomes related to risk perception and behavior than participants in an implemental mindset.

### Methods

A sample of 257 students provided baseline measures on risk perception, readiness to change, and alcohol use. Of those, 64 students with risky alcohol use were randomly allocated to one of two mindset induction conditions–deliberative or implemental mindset. Thereafter, they received the ASSIST-linked Brief Intervention and completed self-report questionnaires on changes in risk perception, alcohol use, and readiness to change at post-intervention and four-week follow-up.

### Results

In contrast to our hypotheses, the four-weeks follow-up revealed that participants in the implemental mindset consumed significantly less alcohol than participants in a deliberative mindset did. The former decreased and the latter increased their alcohol intake; resistance to the brief intervention was stronger in the latter condition. However, neither deliberative nor implemental mindset participants showed any changes in risk perceptions or in their readiness to change alcohol consumption.

**Data Availability Statement:** All relevant data are available in PsychArchives: (https://www.psycharchives.org/handle/20.500.12034/3067) DOI: (https://doi.org/10.23668/psycharchives.3452)

**Funding:** The study was funded by the German Research Foundation, www.dfg.de/en/, Project FOR2374; Subprojects awarded to P.M.G. & L.K. (GO 387/18-1) and M.Odenwald (OD113/2-1). The funders had no role in study design, data collection and analysis, decision to publish, or preparation of the manuscript.

**Competing interests:** The authors have declared that no competing interests exist.

## Conclusions

These findings suggest that mindset induction is a powerful moderator of the effects of the ASSIST-linked Brief Intervention. We argue that systematic research on mindset effects on brief intervention techniques aimed to reduce risky alcohol use is highly needed in order to identify the processes involved with commitment and resistance being the main candidates.

## Introduction

Consuming alcohol in risky or hazardous amounts is common within different populations, but university students in particular represent a high-risk group. They are more likely to drink alcohol compared to non-college groups of the same age [1, 2] and more likely to experience the negative consequences of their drinking patterns [3]. Negative consequences of risky alcohol drinking in young age range from drunk driving, starting physical fights, unsafe sex, academic difficulties to suicidal acts, developing alcohol dependence, heart problems, or cancer [4, 5]. Alcohol consumption is common and widespread among German students, with 37% of them drinking alcohol at least once a week in the past 12 months and 42% reporting binge drinking in the last 30 days [6]. For risky alcohol use, the estimated prevalence rates among students range from 20 to 30% [7, 8]. In Germany, risky drinking is defined as an average daily consumption of more than 12 g pure ethanol in women and 24 g in men; 24 g corresponds to about 0,5–0,6 liter beer or 0,25–0,3 liter wine [9]. Binge drinking is defined as consuming approximately 40–60 g ethanol for women (60–70 g for men) on a single occasion [10].

Although the consequences of hazardous alcohol use are well known, the discrepancy between knowledge of such negative consequences and exhibiting actual risky drinking behavior is widespread [11, 12]. Research on risk perception has attempted to explain the discrepancy between awareness of personal risks and risky alcohol use. For example, Wild and colleagues [13] observed a tendency for optimistic underestimation of a personal experience of harm relative to comparable peers in at-risk drinking students, whereas students with low alcohol use showed no such optimistic bias. Health theories suggest risk perception to be a key factor when it comes to predicting preventive behavior [14, 15].

Screening and Brief Intervention (SBI) is a preventive approach with proven effectiveness to reduce hazardous alcohol consumption that usually consists of an initial assessment, the feedback of the respective results, and additional short interventions; it can be delivered by professionals of different training levels [16, 17]. Although SBIs targeting college students are successful in reducing alcohol consumption and related negative consequences for up to four years afterwards [18], the effect sizes are quite small ($d_+$s = 0.07–0.14) when the interventions are compared to control groups. SBIs often incorporate elements that belong to the FRAMES model [19]; in this model personalized feedback is a central element. For instance, Miller et al. [20] report a significant reduction in drinking among college students for feedback interventions. Similarly, feedback as part of a brief alcohol intervention has proven effective for the prevention of alcohol misuse among first year students [18] and for the reduction of drinking among heavy alcohol consuming college students [21]. Decisional balance is another technique that is frequently used and effective as an additional part of SBIs for college students [20]. A meta-analysis was able to show that most interventions with significant effects on college drinking were delivered face-to-face by skilled professionals while data on the length of the intervention were inconclusive [22]. It is generally accepted that processes like resistance and reactance (e.g., a client rejects the intervention or the counselor) are related to reduced or

lacking effects of interventions [23]. Several studies made this observation with complex substance use disorder interventions [24] as well as brief alcohol advice [25]. Thus, interventions for heavy college drinkers need to be designed to minimize resistance [26].

A theoretical framework to study decisional processes related to behavior change is the mindset theory of action phases [27]. According to this theory, different types of information processing are activated during the different stages of decision making and goal pursuit. Furthermore, it suggests that in the predecisional stage, when facing the task to select suitable and feasible goals and deliberating the pros and cons of specific alternatives, a deliberative mindset is activated which is characterized by open-mindedness for processing new information [28, 29], an impartial processing of information [30], and a realistic view of control [31]. Once the decision is made and the task is to plan the implementation of the goal, an implemental mindset is activated which is characterized by mainly the opposite features: closed-mindedness by ignoring peripheral information [28, 32], partial processing of desirability-related information by preferred thinking about pros over cons [30], and optimistic beliefs about control and feasibility [31, 33]. Moreover, Keller and Gollwitzer [34] observed that asking participants to deliberate the pros and cons of an unresolved personal problem (i.e., activating a deliberative mindset) versus asking people to plan the implementation of a chosen project (i.e., activating an implemental mindset) leads to more realistic risk perceptions and less risk-taking behavior.

Knowing that deliberative versus implemental mindsets facilitate open-mindedness and closed-mindedness, respectively, and that clients' resistance is problematic for the effectiveness of any intervention, we assumed that being in a deliberative mindset would enhance the openness toward and reduce resistance to individualized alcohol risk feedback as it is part of SBIs. Additionally, we assumed that a deliberative mindset is associated with an increased risk perception and decreased risk taking. The present study thus scrutinized the impact of mindset induction on personalized alcohol risk feedback by inducing the mindset right before an SBI. We investigated whether an experimentally induced deliberative mindset translates into increased effects of SBI aimed to reduce risky alcohol use. More specifically, we hypothesized that activating a deliberative mindset versus an implemental mindset could enhance the effectiveness of an alcohol SBI within university students resulting in increased alcohol risk perception, increased readiness to change, and decreased alcohol use.

## Materials and methods

### Procedure and design

This randomized controlled pilot intervention study involved university students with risky alcohol use. It consisted of three sessions (t0, t1 and t2) conducted at a university-based research lab: At t0, participants were screened for hazardous alcohol consumption and answered baseline questionnaires on risk perception and readiness to change alcohol use. Inclusion criteria were current student status and risky alcohol use (past year). Those who qualified were then invited to the second assessment (t1) and were randomly assigned to one of two double-blind experimental conditions, in which one of two mindsets (deliberative vs. implemental) was experimentally induced (see below).

One researcher who did not participate in the provision of the brief intervention (LK) implemented the random assignment. We used an online tool to generate a random allocation sequence using blocks of six random numbers of which three corresponded to each mindset. In the order of their enrollment via an online platform, participants were assigned to IDs and the predefined allocation sequence. For each participant, the allocated mindset manipulation was put into a manila envelope that had a post-it note with the participant's ID on its cover. The experimenters then gave each participant the manila envelope with their ID on it and left

the room before participants opened it and entered the room only after participants put the mindset manipulation back into the envelope. A cover story was used that suggested that the mindset induction was unrelated to the rest of the study. More specifically, participants were told that the experimenter needed to prepare for the upcoming part of the experiment and that the participant could use this time by completing a questionnaire. This questionnaire (i.e., the mindset manipulation) had its own informed consent and stated that it was designed by another group of researchers (i.e., the social psychology and motivation group).

After the mindset induction all participants received the ASSIST-linked Brief Intervention. Thereafter, participants answered self-report questionnaires (alcohol consumption, risk perception, and readiness to change) and participants' resistance (shown during the brief intervention) was rated by the counselors. Four weeks later, a follow-up assessment (t2) took place during which alcohol consumption, risk perception, and readiness to change were measured again. Primary outcome measures were changes in alcohol-related risk perception and alcohol use, secondary outcome measure was readiness to change. All participants were thoroughly debriefed at the end of the study. The trial started in the winter term 2017/18, recruitment was originally planned for two subsequent semesters between November 2017 and October 2018 and t2 assessments were planned to be terminated before the end of the teaching term.

Because no research has ever studied mindset induction effects on brief interventions before we originally estimated that a sample size of $N = 100$ would be required to achieve a power of .8 in a rmANOVA (time * group interaction effect, i.e., 2 * 3) assuming a medium effect size (eta squared = 0.09) and alpha = 0.05. Because of expected dropout, we originally planned to recruit up to 120 participants. We did not include a non-mindset control group as originally planned due to restricted resources. We decided to stop further recruitment after an interim analysis in February 2018 revealed the real effect sizes and an unexpected increase of alcohol use in one group.

The study protocol was approved by the Institutional Review Board of the University of Konstanz, Germany; the trial registry number is NCT03338491 (www.ClinicalTrials.gov). According to the IRB approval participants of the screening gave informed consent by clicking the respective button in the experiment management system. All participants of the intervention study gave written informed consent. All intervention study participants were fully informed about the study after completing the follow-up assessment.

## Participants

Two hundred fifty-seven students (72% female) of a German university were recruited via an experiment management system to participate in an online survey (t0) which included the screening for risky alcohol use. Of them, $N = 113$ students (i.e., 44%) exhibited hazardous alcohol consumption and were invited to the intervention session (t1). From this invited sample, $n = 66$ participated and were randomly assigned to one of two experimental conditions (mindsets: deliberative vs. implemental before receiving a brief intervention to reduce alcohol use). On average, participants (68% female) were 20.9 years of age ($SD = 2.4$; min = 18, max = 30). From this sample, $n = 64$ were reached at the four-weeks follow-up. The two participants who missed their t2 appointment did not answer further invitations. Fig 1 summarizes the participant flow.

## Measures and instruments

**Screening for hazardous alcohol use.** At t0 hazardous alcohol use was assessed by the Alcohol Use Disorder Identification Test [AUDIT; 35], a reliable and valid measure for risky

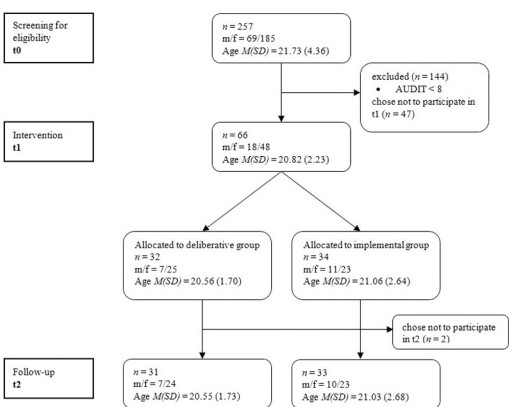

**Fig 1. Flow chart.**

alcohol use [36]. According to the suggestions of the WHO [35], participants with an AUDIT score of eight or more were included into the study.

**Alcohol use and risk-taking behavior.** The timeline follow-back method [TLFB; 37] was used to quantify actual alcohol use in the 28 days before t1 and t2, respectively. The TLFB is a reliable and valid calendar-based measure of daily alcohol use [38]. Via self-report, participants estimated their daily consumption retrospectively for the last 28 days before the assessment. Alcohol consumption was measured in standard drinks and the total number of standard drinks was used as main dependent variable.

**Readiness to change and risk perception.** The German version of the Stages of Change Readiness and Treatment Eagerness Scale (SOCRATES) [39] [40] is a validated and reliable questionnaire for measuring the readiness to change problem drinking. The SOCRATES includes three subscales: Recognition, Ambivalence, and Taking Steps. Additionally, participants filled out the Precontemplation subscale of the validated German short version of the University of Rhode Island Change Assessments [URICA; VSS-k; 41]. We used the Domain-Specific Risk-Taking Scale [DOSPERT; 42] [43] in its validated German version to assess general risk perceptions. The DOSPERT consists of 30 risks that have to be rated on the willingness to take each risk (e.g., "How likely is it that you are going camping in the wilderness?"). Furthermore, we used the German questionnaire "Fragebogen zur alkoholbezogenen Risikowahrnehmung" [FAR; 11] to capture alcohol-related risk perceptions. The FAR consists of 20 items measuring alcohol-related risk perceptions in four domains: perceived personal vulnerability, peer vulnerability, affective risk perception, and precaution effectiveness. Each domain consists of five items evaluated on a five-point Likert scale. The SOCRATES, the URICA Precontemplation Scale, the FAR, and the DOSPERT were filled out at t0, t1, and t2. For SOCRATES and the URICA Precontemplation Scale, we modified the original Likert answer scales into visual analogue scales to prevent response biases due to repeated assessments (i.e. respondents remember their previous answers); we report percentage scores with 100% representing the highest possible value.

**Resistance.** After the ASSIST-linked Brief Intervention, the counselors rated how resistant they perceived the participants to be during the interview on a five-point answer scale (1 = "not at all" to 5 = "extremely high") addressing the question "How much resistance did the participant show during the intervention?". We included this rating during the last half of data collection for t1, which it was obtained for only a subset of our sample (n = 26) in order to perform an additional explanatory analysis. Because there was only one counselor present for each intervention session, there were no multiple ratings of resistance per participant.

## Interventions

**Mindset manipulation.** In research on the mindset theory of action phases, the deliberative and implemental mindsets are typically induced by a procedure developed by Gollwitzer and colleagues [overview by 44]. Both deliberative and implemental mindsets are assumed to carry over to different unrelated tasks, which the participants are asked to perform afterward. In our study, mindsets were activated as described in detail by Keller and Gollwitzer [34]. Participants in the deliberative mindset condition were instructed to name an unresolved interpersonal problem of the type "Should I leave it as is or should I try to make a change?", occupying their mind for which they had not made any decision yet whether to take action or not. They were asked to name their problem in the format of "Should I do . . . or not?". After that, participants were instructed to weigh positive and negative, immediate and long-term consequences of making or not making a change. In contrast, participants in the implemental mindset condition had to name an interpersonal project that they already had decided to resolve but had not initiated any actions yet. The project should have the form of "I intend to do . . .!". Participants in the implemental mindset condition were then instructed to name five steps necessary for the completion of the project and specify where, when, and how they would implement these steps. We asked for problems/projects from the interpersonal domain, thus preventing participants naming alcohol related problems/projects. As a manipulation check, participants of both conditions were asked to mark their position on a decision timeline, indicating whether they saw themselves before or after making a decision in the selected problem/project; we measured the position on the timeline in cm from "0" (point of making a decision), with negative numbers indicating being before and positive numbers indicating being after the point of making a decision.

**Brief intervention.** The ASSIST-linked Brief Intervention, consisting of the Alcohol, Smoking and Substance Involvement Screening Test (ASSIST) and the associated brief intervention [45], is a standardized SBI that contains a strong personalized feedback element. With eight items, the ASSIST interview assesses the current and lifetime use of alcohol and other substances as well as substance-related symptoms. For each substance category, an individual risk score can be calculated determining a low-, moderate-, or high-risk level. The ASSIST interview achieves good reliability and validity [46]. In the standardized ASSIST-linked Brief Intervention, participants receive feedback on the identified alcohol risk-score as well as information on related individual risks for social and health problems. The ASSIST-linked BI consists of ten steps that are centered on a personalized feedback (Part 1) and a decisional balance exercise (Part 2): 1. Asking clients if they are interested in getting to know their risk scores, 2. provision of personalized feedback, 3. giving advice how to reduce risk, 4. allowing clients to take responsibility for their choices, 5. asking how concerned clients are, 6. balancing good things about alcohol use against 7. the less good things, 8. summarizing and reflecting the clients' statements with emphasis on less good things, 9. asking clients how concerned they are about the less good things, and 10. providing take-home materials (self-help booklet). When delivering the ASSIST-linked BI, motivational interviewing techniques are used to reduce clients' resistance and elicit change talk. The total duration of the ASSIST-linked Brief Intervention was about 30 min and was conducted by postgraduate psychology students who were intensively trained by licensed psychotherapists using a combination of theoretical and practical methods, including role plays and on site supervision.

## Statistical procedures and handling of missing data

One person declined to act on the implemental mindset instructions. All other data for this participant were obtained and thus handled in an intention-to-treat analysis. For replacement

of missing data due to dropout (two participants), the multiple imputation technique was used (Little's MCAR test, $\chi^2$ (22) = 28.24, $p$ = .168). In addition to the reported results below, we analyzed the data excluding these three participants to see if our conclusions would change but they did not.

To test for baseline differences between mindset groups, $\chi^2$-tests were performed for categorical variables and univariate ANOVAs for continuous variables. If Levene tests revealed heterogeneity of variances, Mann-Whitney tests were used instead. For the manipulation check item, a *t*-test was used to check whether participants in the implemental mindset group rated themselves differently from participants in the deliberative mindset group on the timeline concerning their decision state. To assess the effects of the mindset induction, we subjected the variables of interest (i.e., risk perception, readiness to change, precontemplation) to a 2 between (Mindset: deliberative versus implemental) x 3 within (Time: t0, t1, t2) ANOVA. The Greenhouse-Geisser adjustment was used to correct for violations of sphericity. We also utilized the same type of ANOVA to compare the two mindset groups concerning the change of the total alcohol use from t1 to t2. We chose ANOVAs as many findings speak for the robustness of the analysis of variance concerning violated assumptions, such as non-normally distributed data [47, 48]. Homogeneity of variances was asserted using Levene's test that showed that equal variances could be assumed, except for the variable of precontemplation. To test for differences in the resistance rating between the two mindset groups, a Mann-Whitney U test was performed. Results with a Type I error rate of $p < 0.05$ in two-sided tests were considered statistically significant. Analyses were performed using SPSS version 25.

## Results

All initially included participants were included into analysis.

### Baseline characteristics and manipulation check

The two experimental groups did not differ in baseline characteristics: gender distribution, $\chi^2$ (1) = 0.91, $p$ = .34, age, $F(1, 64)$ = 0.81, $p$ = .37, or pre-intervention AUDIT scores nor alcohol consumption, $Fs(1, 62) < 1$. On average, participants drank around 39.4 ($SD$ = 23.1) standard drinks in the month before t1. Baseline risk perceptions (general and alcohol-related), readiness to change, and precontemplation were similar in both groups as described in Table 1, all $p$s ≥ .135.

The manipulation check indicated that the mindset induction was successful. The two groups differed on the decision timeline, $t(58.0)$ = 2.41, $p$ = .019, with participants in the deliberative mindset condition indicating to be before the decision ($Med$ = - 2.4 cm) and participants in the implemental mindset condition indicating to be right at the decision ($Med$ = 0.0 cm).

### Mindset effects on SBI outcomes

**Risk perception.** To test our hypothesis that participants differ in their general risk perception over time depending on whether they are in a deliberative versus an implemental mindset, a repeated-measures ANOVA was conducted. It revealed no significant main effect for mindset condition, $F(1, 64)$ = 0.87, $p$ = .356 $\eta_p^2$ = .013, and no significant interaction between mindset condition and time, $F(1.7, 107.7)$ = 0.66, $p$ = .496, $\eta_p^2$ = .010, although there was a significant main effect for time, $F(1.7, 107.7)$ = 6.69, $p$ = .003, $\eta_p^2$ = .095. Participants increased their general risk perception as measured by the DOSPERT over the course of the three sessions (see Table 2). Furthermore, general risk perception as measured in the DOSPERT correlated with alcohol consumption both at t1 and t2, $r(n = 66)$ = .29, $p$ = .017, and r

**Table 1. Baseline characteristics of the sample.** We report M (SD) or N (%).

|  | Total Group (*N* = 66) | Implemental Mindset (*N* = 34) | Deliberative Mindset (*N* = 32) | $p^5$ |
|---|---|---|---|---|
| Age | 20.8 (2.2) | 21.1 (2.6) | 20.6 (1.7) | .371 |
| Gender |  |  |  |  |
| female | 48 (72.7%) | 23 (67.6%) | 25 (78,1%) | .339 |
| male | 18 (27.3%) | 11 (32.4%) | 7 (21.9%) |  |
| AUDIT | 12.11 (3.7) | 11.7 (3.9) | 12.5 (3.3) | .363 |
| DOSPERT | 3.7 (0.7) | 3.8 (0.8) | 3.6 (0.7) | .498 |
| FAR-PPV[1] | 1.8 (0.5) | 1.8 (0.5) | 1.8 (0.5) | .908 |
| FAR-PV[2] | 2.2 (0.7) | 2.2 (0.8) | 2.2 (0.7) | .820 |
| FAR-ARP[3] | 3.1 (1.0) | 3.1 (1.1) | 3.0 (0.9) | .720 |
| FAR-PE[4] | 1.6 (0.5) | 1.6 (0.6) | 1.5 (0.4) | .638 |
| SOC-Recognition | 111.3 (124.5) | 123.3 (149.8) | 98.6 (91.2) | .425 |
| SOC-Ambivalence | 108.4(92.4) | 111.9 (96.2) | 104.7 (89.5) | .754 |
| SOC-Taking Steps | 184.7 (187.8) | 218.3 (200.2) | 149.1 (169.6) | .135 |
| URICA Precontemplation | 36.6 (18.2) | 34.7 (20.5) | 38.6 (15.4) | .166 |

[1] FAR subscale perceived personal vulnerability

[2] FAR subscale peer vulnerability

[3] FAR subscale affective risk perception

[4] FAR subscale precaution effectiveness

[5]Results of the comparison between mindset conditions

(n = 66) = .38, p = .001, respectively (see S1 Table). Repeating the rmANOVA with gender as an additional IV revealed no gender effects at all.

Testing whether alcohol-related risk perceptions changed depending on the mindset condition, repeated-measures ANOVAs revealed no significant interactions between mindset condition and time, all *F*s $\leq$ 1.12, all *p*s $\geq$ .329, all $\eta_p^2$s $\leq$ .017, nor significant main effects, all *F*s $\leq$ 1.48, all *p*s $\geq$ .232, all $\eta_p^2$s $\leq$ .023, in all four FAR domains. Including gender into the rmANOVA revealed a significant two-way interaction between time and gender for the personal vulnerability domain but no interactions with mindset condition. It also revealed an interaction between gender and mindset condition for the affective risk perception domain. There were no gender effects found for the other two domains. However, the personal vulnerability domain correlated with alcohol consumption both at t1 and t2, *r*(*n* = 66) = .25, *p* = .040, and *r*(*n* = 66) = .31, *p* = .013 (see S1 Table).

**Alcohol use.** When comparing the two mindset conditions with respect to the total alcohol consumption over time (t1, t2), we observed a significant decrease of alcohol consumption for the implemental mindset condition and an increase in the deliberative mindset condition. Hence, the ANOVA showed a significant main effect for the mindset condition, F(1, 64) = 5.72, p = .020, $\eta_p^2$ = .082, no effect of time, F(1, 64) = 0.09, p = .768, $\eta_p^2$ = .001, but a significant interaction between mindset condition and time F(1, 64) = 6.74, p = .012, $\eta_p^2$ = .095. Post-hoc paired t-tests revealed that participants in the implemental mindset condition exhibited a trend in reducing their alcohol intake by an average of almost 6 standard drinks between t1 and t2, t(33) = 1.56, p = .129, while participants in the deliberative mindset condition significantly increased their alcohol intake on average by more than 7 standard drinks between t1 and t2, t(31) = 2.16, p = .038. Alcohol intake did not differ between mindset conditions at t1, F(1, 64) = 1.80, p = .185, $\eta_p^2$ = .027, but did differ at t2, F(1, 64) = 8.39, p = .005, $\eta_p^2$ = .116. These findings are illustrated in Fig 2. Repeating this analysis with gender as an additional factor revealed no significant gender effects.

**Table 2. Outcome variables.** We report M (SD).

| Variable | Group | Baseline (t0) | Post (t1) | Follow-up (t2) |
|---|---|---|---|---|
| Alcohol Standard Units[1] | Implemental Mindset | - | 35.71 (22.73) | 29.81 (24.09) |
| | Deliberative Mindset | - | 43.29 (23.22) | 50.70 (33.96) |
| DOSPERT | Implemental Mindset | 3.75 (0.80) | 3.81 (0.73) | 3.97 (0.68) |
| | Deliberative Mindset | 3.63 (0.69) | 3.70 (0.60) | 3.76 (0.53) |
| FAR-PPV[2] | Implemental Mindset | 1.79 (0.51) | 1.81 (0.48) | 1.86 (0.58) |
| | Deliberative Mindset | 1.80 (0.55) | 1.87 (0.48) | 1.87 (0.43) |
| FAR-PV[3] | Implemental Mindset | 2.18 (0.79) | 2.22 (0.68) | 2.09 (0.74) |
| | Deliberative Mindset | 2.22 (0.71) | 2.30 (0.56) | 2.19 (0.61) |
| FAR-ARP[4] | Implemental Mindset | 3.11 (1.08) | 3.16 (0.96) | 3.06 (0.94) |
| | Deliberative Mindset | 3.02 (0.93) | 3.33 (0.79) | 3.32 (0.99) |
| FAR-PE[5] | Implemental Mindset | 1.64 (0.62) | 1.54 (0.43) | 1.65 (0.49) |
| | Deliberative Mindset | 1.51 (0.37) | 1.53 (0.44) | 1.56 (0.38) |
| SOC-Recognition | Implemental Mindset | 123.32 (149.77) | 134.53 (128.64) | 124.31 (145.89) |
| | Deliberative Mindset | 98.63 (91.16) | 106.69 (95.77) | 101.85 (91.30) |
| SOC-Ambivalence | Implemental Mindset | 111.85 (96.17) | 107.91 (74.65) | 99.87 (91.48) |
| | Deliberative Mindset | 104.66 (89.53) | 102.75 (84.61) | 106.62 (83.64) |
| SOC-Taking Steps | Implemental Mindset | 218.32 (200.20) | 244.21 (190.21) | 234.93 (191.02) |
| | Deliberative Mindset | 149.06 (169.56) | 183.53 (154.32) | 176.73 (146.96) |
| URICA Precontemplation | Implemental Mindset | 34.66 (20.45) | 28.65 (18.61) | 28.34 (18.52) |
| | Deliberative Mindset | 38.59 (15.43) | 31.22 (15.97) | 27.55 (15.17) |

[1] Alcohol Standard Units consumed in the past 24 days

[2] FAR subscale perceived personal vulnerability

[3] FAR subscale peer vulnerability

[4] FAR subscale affective risk perception

[5] FAR subscale precaution effectiveness

**Readiness to change.** Exploring readiness to change, a repeated-measures ANOVA showed no statistically significant interaction between time and mindset group, all $Fs \leq 0.49$, all $ps \geq .577$, all $\eta_p^2 s \leq .008$, nor significant main effects, all $Fs \leq 2.44$, all $ps \geq .118$, all $\eta_p^2 s \leq .037$, in all three subscales. Furthermore subjecting Precontemplation to a repeated-measures ANOVA, the results revealed no significant main effect for mindset condition, $F(1, 64) = 0.27$, $p = .608$, $\eta_p^2 = .004$, but a significant main effect of time, $F(2, 128) = 11.01$, $p < .001$, $\eta_p^2 = .147$,

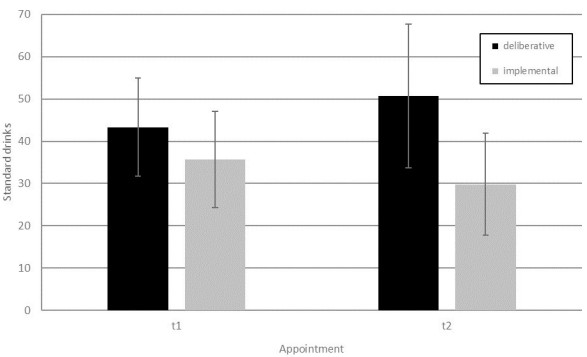

**Fig 2. Amount of alcoholic standard drinks in the 4 weeks before and after the intervention.** We report means and standard deviation.

indicating that scores on the Precontemplation scale decreased over the course of the three sessions of our experiment. However, the interaction between mindset condition and time did not reach statistical significance, $F(2, 128) = 0.79$, $p = .457$, $\eta_p^2 = .012$. In Table 2, we provide an overview of the average scores for each of the outcome variables. Including gender in the rmANOVAs revealed no interactions between mindset and gender; all three SOCRATES subscales showed an interaction between time and gender, and Precontemplation showed a gender main effect.

**Resistance (exploratory analysis).** We then compared resistance as rated by the counselors between the deliberative and implemental mindset conditions and found that participants in the deliberative mindset condition showed more resistance to the intervention than participants in the implemental mindset condition, U = 38.00, p = .016.

## Discussion

In the present study, we investigated the influence of a mindset induction on the effectiveness of a standardized SBI protocol, the ASSIST-linked BI, containing a personalized alcohol feedback and a decisional balance exercise to reduce risky alcohol use among university students. We found that activating an implemental mindset in participants before the intervention took place showed a reduction of alcohol use in the subsequent four weeks after the intervention, while the participants who had been placed in a deliberative mindset actually showed an increase in drinking. While this was independent of the participants' gender, it is in contrast to our hypotheses: We had expected that the induction of a deliberative versus an implemental mindset would enhance the acceptance of the alcohol feedback and, thus, increase participants' risk perceptions and readiness to change, leading to reductions in alcohol consumption. Contrary to our hypotheses, risk perception and readiness to change remained unchanged and participants in the implemental mindset condition showed reduced risk behavior compared to participants in the deliberative condition. Also contrary to our assumptions, implemental mindset participants showed less resistance during the brief intervention compared to deliberative mindset participants, as rated by their respective counselor who was blind to the participants' mindset conditions after delivering the SBI. Thus, our empirical results demonstrate mindset effects that are opposite to our hypotheses. Still, they hint at mindset induction as a potentially powerful intervention tool, and they raise questions regarding the mechanisms and cognitive processes underlying our results.

But how can we explain our results? The manipulation check suggests that the deliberative and implemental mindsets were induced as intended. Still, resistance occurred to a higher extent during the SBI in the deliberative compared to the implemental mindset group, which was unexpected. What could be the reasons for this unexpected occurrence of resistance? One possible explanation relates to the components of the intervention used. The ASSIST-BI does entail two components, a. the personalized feedback and b. the decisional balance exercise. With respect to the first component, we see no reason why the deliberative mindset participants did not benefit from the open-mindedness associated with the deliberative mindset in their processing of the personalized feedback. In related mindset studies, deliberative mindset participants were indeed found to effectively adjust their risk perception after negative feedback more so than implemental mindset participants [34]. Please note, however, that because of how we designed our intervention sessions and measured resistance, we cannot provide inter-rater reliability as only one counselor was present at each intervention session.

The second component of the ASSIST-BI, the decisional balance exercise, might therefore be more relevant to explaining our unexpected results in the deliberative mindset group. In their review on decisional balance procedures, Miller and Rose [49] conclude that employing

this technique with undecided individuals will decrease commitment for change because the benefits of the status quo are brought to one's attention and "sustain talk" is elicited. They refer to a number of studies that report this effect: For example, in a series of experimental studies with university students, Nenkov and Gollwitzer [50] showed that predecisional individuals reduced their commitment to pursuing a given goal after they had participated in a decisional balance exercise regarding this goal. In a clinical sample with heavy college drinkers, Carey et al. [51] report that a basic Brief Motivational Intervention (BMI) consisting of personalized feedback of alcohol risk levels and psychoeducation had better drinking and risk outcomes than an enhanced BMI in which a decisional balance exercise was added to the basic module. Also Krigel et al. [52] showed that a decisional balance exercise did not increase outcomes in student smokers not intending to quit. The specific challenges of a decisional balance exercise with predecisional clients that need to be met by therapists are highlighted by Gaume et al. [53]. These authors evaluated a brief MI aimed to reduce alcohol use among young heavy drinkers and found that inexperienced therapists provoked an increase in drinking when performing motivational interviewing less skillfully than experienced therapists. The studies by Carey at al. and Krigel at al. described above also employed inexperienced therapists; the interventions were either implemented by trained graduate students or basic training level therapists newly trained in motivational interviewing.

In sum, our unexpected results in the deliberative mindset condition may be explained by the following arguments: While the feedback part of the study could have worked in the intended direction, the subsequent decisional balance exercise probably overwrote it with opposing effects. In our decisional balance exercise the counselors also asked about the perceived good aspects of alcohol; this question could have triggered sustain talk that counteracted behavior changes. Our counselors had little motivational interviewing experience and might not have managed to maneuver around sustain talk that counteracted the positive personalized feedback effects. Additionally, the counselors' attempts to control sustain talk may have provoked resistance which further worsened the intervention effects.

With respect to the implemental mindset group, we expected that the feedback part of the intervention was received with less openness. With respect to the decisional balance exercise part, Nenkov and Gollwitzer [50] and Miller and Rose [49] report that a decisional balance exercise engaged in by postdecisional individuals strengthens goal commitment and respective goal-directed behavior. The authors explain this phenomenon by pointing to postdecisional defensiveness [50] and efforts to reduce cognitive dissonance [49], leading to selectively favoring arguments in support of the prior taken decision. Unfortunately, we did not measure commitment itself but only outcomes that implied heightened commitment. However, in our study, the prior taken decision used to induce an implemental mindset was not related to the question of whether or not to reduce alcohol use. Therefore, the critical question is, how could it happen that alcohol use decreased even though the implemental mindset was induced by planning the implementation of a completely unrelated decision? Therefore, it cannot be postdecisional defensiveness or attempts to reduce cognitive dissonance, which would only make sense when the decision and the respective subsequent decisional balancing exercise are targeting the same decision problem. Obviously, the decisional balancing exercise in our implemental mindset group must have evoked different cognitive mechanisms, all to be explored in future studies. These studies might want to explore whether the implemental mindset is implicitly carried over to a question not yet decided, and that information on pros and cons of alcohol use is now processed as if a decision has already been made. Supportive evidence for this possibility comes from our follow-up assessment where we directly asked our participants whether they intended to reduce alcohol use right after the intervention or not; the majority answered "no", without differences between mindset groups ($p = .230$). Additional support comes from the observation

that participants in the implemental mindset showed behavior change without the expected change in the underlying motivational factors, readiness to change and risk perceptions. In addition, a further possibility is that the implemental mindset leads to an implicit decision regarding the question at hand (i.e., "reduce alcohol use or leave it as it is?"), a cognitive process of „jumping to decisions"(analogous to „jumping to conclusions").

In sum, our unexpected results raise a number of new questions. An experimental approach to answer these questions about the processes elicited by the two distinct components of the ASSIST-BI and their differential interaction with deliberative and implemental mindsets would require a 2 (mindsets: deliberative vs. implemental) x 2 (component: feedback vs. decisional balance exercise) x 2 (level of counselors' motivational interviewing experience) with separate measures of resistance and commitment ratings as well as subsequent behavioral change. It is hypothesized that among the clients of inexperienced counselors deliberative mindset participants would show low resistance during personalized feedback and high resistance after a decisional balance exercise, and the opposite pattern for commitment. A standardized training would help to implement the different MI skill levels of therapists, e.g. a training for using the different methods to evoke change talk or to avoid sustain talk. Implemental mindset participants are expected to show the opposite pattern to the deliberative mindset participants for resistance and commitment after a decisional balance exercise irrespective of counselors' experiences with motivational interviewing; it remains unclear how this group would respond to a personalized feedback procedure. Furthermore, the participants' alcohol use should reflect the expected finings for resistance and commitment.

We also found that risk perception and readiness to change were not influenced by the brief intervention in both mindset groups. This is in line with a recent systematic review where both constructs did not emerge as mediators of intervention effects regarding the reduction of college student drinking [54]. But although no support was found for our hypotheses that the specific mindset during an intervention has an influence on the change of the variables risk perception and readiness to change alcohol consumption, it does not necessarily imply that there are no mindset and intervention effects on these variables. It would be premature however to conclude that these variables were unresponsive as we did not study their trajectories. We measured them at baseline, just after the intervention and follow-up one month later. Based on the risk reappraisal hypothesis [55] one would expect that after a behavior change, risk perception is adapted; in the case of implemented alcohol use reduction, alcohol risk perception (especially the domain perceived personal vulnerability) should eventually decrease. In our study, the timing of assessment of risk perception might not have captured this dynamic. In order to measure trajectories of risk perception, a more frequent measurement in everyday life would be necessary, such as ecological momentary assessment.

Several limitations of the present study should be noted. The major limitation are the missing no-mindset and no-intervention control groups. Thus, the reduction of alcohol consumption after the brief intervention cannot be clearly attributed to the induction of an implemental mindset compared to a deliberative mindset. Also, we cannot say whether mindset induction alone without brief intervention would already affect alcohol use. The present results need to be replicated in a study with a more complete design that contains an additional control group without any mindset induction, and control groups which receive no brief intervention after the deliberative or implemental mindset inductions. A further limitation is that the counselors were no experienced therapists trained in motivational interviewing. Instead, we used a manualized version, the ASSIST-linked Brief Intervention, due to restricted resources. Moreover, all outcomes were assessed by self-reports which are vulnerable to social desirability [56]. A final limitation is the non-representative sample consisting mostly of female students in their first semester, which was due to our recruiting strategy.

## Conclusion

In the present study, deliberative versus implemental mindsets were induced before participants received a standardized SBI containing personalized feedback and a decisional balance exercise to reduce risky alcohol consumption. Alcohol use reduced clearly in the implemental mindset group in the four weeks after the intervention, while it increased in the deliberative mindset group. Participants showed no meaningful changes in readiness to change and alcohol-related risk perceptions. The present study offers useful insights into drinking behavior in a student sample of risky drinkers and into the mechanisms related to the effectiveness of brief interventions on risky drinking.

## Supporting information

**S1 Table. Inter-correlation of variables at baseline.**
(DOCX)

**S1 Checklist.**
(DOC)

**S1 File.**
(PDF)

## Acknowledgments

We thank Brigitte Rockstroh and Daniela Mier for their support.

## Author Contributions

**Conceptualization:** Natascha Büchele, Lucas Keller, Peter M. Gollwitzer, Michael Odenwald.

**Data curation:** Natascha Büchele, Lucas Keller.

**Formal analysis:** Natascha Büchele, Lucas Keller, Michael Odenwald.

**Funding acquisition:** Lucas Keller, Peter M. Gollwitzer, Michael Odenwald.

**Investigation:** Natascha Büchele, Lucas Keller, Anja C. Zeller, Freya Schrietter, Julia Treiber, Michael Odenwald.

**Methodology:** Natascha Büchele, Lucas Keller, Peter M. Gollwitzer, Michael Odenwald.

**Project administration:** Natascha Büchele, Lucas Keller, Anja C. Zeller, Freya Schrietter, Julia Treiber, Michael Odenwald.

**Resources:** Natascha Büchele, Lucas Keller, Peter M. Gollwitzer, Michael Odenwald.

**Software:** Lucas Keller.

**Supervision:** Natascha Büchele, Lucas Keller, Peter M. Gollwitzer, Michael Odenwald.

**Validation:** Natascha Büchele, Lucas Keller, Michael Odenwald.

**Writing – original draft:** Natascha Büchele, Michael Odenwald.

**Writing – review & editing:** Natascha Büchele, Lucas Keller, Anja C. Zeller, Freya Schrietter, Julia Treiber, Peter M. Gollwitzer, Michael Odenwald.

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
