## [Decision Letter · Decision Letter 0]

19 Dec 2019

PONE-D-19-18841

The Effects of Mindset Induction on a Brief Intervention to Reduce Alcohol Use and Increase Risk Perception among University Students: A Pilot Randomized Controlled Trial

PLOS ONE

Dear Dr. Odenwald,

Thank you for submitting your manuscript to PLOS ONE. After careful consideration, we feel that it has merit but does not fully meet PLOS ONE’s publication criteria as it currently stands. Therefore, we invite you to submit a revised version of the manuscript that addresses the points raised during the review process.

This manuscript has been reviewed by two independent reviewers; their comments are below. The reviewers are largely positive about the work but have requested several points of clairification of the methods and results, as well as attention to English lanugage grammar and usage. Please revise your manuscript to address the reviewers’ comments, as well as any journal requests.

We would appreciate receiving your revised manuscript by Feb 01 2020 11:59PM. To enhance the reproducibility of your results, we recommend that if applicable you deposit your laboratory protocols in protocols.io, where a protocol can be assigned its own identifier (DOI) such that it can be cited independently in the future. For instructions see: http://journals.plos.org/plosone/s/submission-guidelines#loc-laboratory-protocols

We look forward to receiving your revised manuscript.

Kind regards,

Vanessa Carels

Staff Editor

PLOS ONE

Journal Requirements:

2. Please provide additional details regarding participant consent.

In the ethics statement in the Methods and online submission information, please ensure that you have specified (a) whether consent was informed and (b) what type you obtained (for instance, written or verbal, and if verbal, how it was documented and witnessed).

If your study included minors, state whether you obtained consent from parents or guardians. If the need for consent was waived by the ethics committee, please include this information.

Reviewers' comments:

Reviewer's Responses to Questions

**Comments to the Author**

1. Is the manuscript technically sound, and do the data support the conclusions?

Reviewer #1: Yes

Reviewer #2: Yes

2. Has the statistical analysis been performed appropriately and rigorously? 

Reviewer #1: I Don't Know

Reviewer #2: Yes

3. Have the authors made all data underlying the findings in their manuscript fully available?

Reviewer #1: Yes

Reviewer #2: No

4. Is the manuscript presented in an intelligible fashion and written in standard English?

Reviewer #1: Yes

Reviewer #2: No

5. Review Comments to the Author

Reviewer #1: Thank you for your paper, it is an interesting read, and important for the field. It is a well written paper, and I have the following minor comments to make:

Abstract:

Line 23, "showed" rather than "did show"

Introduction:

Line 45, "perception is a key factor" rather than "as a key factor"

SBIs vary markedly in their components and length. Can citations be included with specific details as to which have been found most effective, and in particular for students as your group of interest.

Method:

Lines 116-117: At what time point was recruitment stopped? By how much did the alcohol use increase, hence halting recruitment?

Line 180: should it read "I intend to do!"?

Line 204: who trained the students? Trained in what respect?

Reviewer #2: Review of PONE D-19-18841

This is an interesting and elegantly designed study yielding valuable results concerning the unexpected effects of induction of deliberative versus implemental mindsets among university students with risky drinking, who all received the ASSIST brief intervention following the induction. The manuscript is generally clear but some clarifications are needed and, mainly, a thorough English-language review. See general and specific comments below.

General comments

1. The abstract needs to be clarified somewhat, see below. ´

2. The definition of risky drinking differs between men and women, as indicated in the first paragraph of the introduction. The authors do not report any gender-related analyses, however. Please complement the results with information on any gender differences, as well as information on the percentage of participants (by gender) over the risky drinking level at t2; the reviewer’s understanding is that 100% were risky drinkers at t1.

3. The conceptual model linking mindset, risk perception, readiness to change, the intervention consisting of PF and decisional balance, and the alcohol use outcome, would be clearer if a figure were provided, either in the introduction or in the discussion. Indeed, it is not entirely clear why alcohol use was not designated as the sole primary outcome, notwithstanding the authors’ review of earlier research supporting risk perception and readiness to change as moderators of alcohol use outcomes following intervention.

4. In the method section. The mindset induction is clearly described. However, it is stated that “a cover story was used…”. Please explain the rationale for the cover story and describe the instructions given.

5. The measures are adequately described, but it is not clear whether the German versions have been validated or simply translated. Please clarify. Also, it is not clear how resistance was measured by the student counselors, and whether any inter-rater reliability procedure was used (if not, include in limitations in the discussion).

6. The results are also adequately described, but they are not presented in the order of the primary and secondary outcomes specified in the manuscript as well as in the trial registration description. The risk perception and alcohol use outcomes should be presented first, and the readiness to change outcomes should be presented afterwards, as the secondary outcomes.

7. Table A1, showing the t2 study outcomes, should be included in the manuscript as Table 2, not in the Appendix. (Table A2 can remain in the Appendix).

8. The discussion of the deliberative and implemental mindsets as related to the unexpected outcomes on alcohol use is interesting and well-informed. However, the discussion of the implemental mindset results could be revised to include the motivational interviewing (MI) approach described in “Ten strategies for evoking change talk”, namely asking the client to describe previous successful experiences of making a decision to change.

9. Regarding the discussion of future studies, factorial design (LM Collins, Optimization of Behavioral, Biobehavioral, and Biomedical Interventions The Multiphase Optimization Strategy (MOST), 1st ed. 2018. ed. Cham: Springer International Publishing : Imprint: Springer, 2018) could be an ideal way of evaluating the questions posed, in a design evaluating mindset (deliberative vs implemental vs none), MI skills (high vs low), Personal feedback (yes/no) and decisional balance (yes/no). Factorial designs, although complex, identify optimal components to include in interventions (e.g., Crane D, Garnett C, Michie S, West R, Brown J. A smartphone app to reduce excessive alcohol consumption: Identifying the effectiveness of intervention components in a factorial randomised control trial. Scientific reports. 2018;8(1):4384.). A larger sample would be needed, but not as large as a traditional randomized controlled trial.

Specific comments

1. The title is a bit unclear. Perhaps:

Brief intervention to increase risk perception and reduce alcohol use among university students: A pilot randomized controlled trial evaluating the effects of pre-intervention mindset induction

2. The abstract is also imprecise:

• Line 10. Effects, not consequences

• Lines 10-11 very complex sentence, revise. “the openness and thereby the effectiveness…” gives unnecessary information in the abstract and is confusing.

• Line 14. Better outcomes – better than what?

• Line 15. Sample of 257 students, not 256.

• Lines 15-18, revise: “A sample of 257 students provided baseline measures on risk perception, alcohol use and readiness to change, after which 63 students with risky alcohol use were randomly allocated to one of two mindset induction conditions – deliberative or implemental. Then, they received an in-person ASSIST-linked brief inyervention and completed self-report questionnaires on changes in risk perception, alcohol use, and readiness to change at post-intervention and four-week follow-up.”

• Lines 23-24. English: should be “neither….showed any changes in risk perception or in readiness to change…”

• Line 27. “the processes involved” is too vague; specify.

3. Introduction

• Lines 67-68, delete “Whereas”; enough to say “Once the decision is made…”.

4. Method

• Line 92, should be “in which one of two mindets…” to be clear.

• Line 98, delete “just”, add “on the cover of the envelope”.

• Lines 102-104, explain “cover story”, see above.

• Lines 110-111, can be part of the paragraph above.

• Line 113. Should read: “sample size…would be required…”

• Participants – adjust English.

• Line 128. “second appointment” – meaning what? T2?

5. Results

• Lines 253-254, a bit too much information? Personal vulnerability earlier described as “own vulnerability”, keep it consistent. Personal or self better than “own”.

• Lines 261-262. Do not include non-significant results or trends.

6. Discussion

• Lines 323-324. Should read “These authors evaluated a brief MI intervention aimed to reduces alcohol use among young heavy drinkers and found that inexperienced therapists…”

• Line 373. Emerge, not evince.

• Line 378. Not necessarily so that risk perception would decrease after implementing alcohol use reduction. Perhaps personal vulnerability would decrease, but the risk perception should actually have increased, not decreased. Please clarify.

Table 1

• Test statistic not necessary to include.

Although the writing is generally clear, the journal does not provide copy-editing and the English should be reviewed by a native speaker with scientific editing skills.

6. PLOS authors have the option to publish the peer review history of their article (what does this mean?). If published, this will include your full peer review and any attached files.

Reviewer #1: No

Reviewer #2: Yes: Anne H Berman

---

## [Author Response · Author response to Decision Letter 0]

15 Feb 2020

Please find the detailed responses to reviewers' comments and suggestions in the uploaded document.

---

## [Decision Letter · Decision Letter 1]

8 Jun 2020

PONE-D-19-18841R1

The effects of pre-intervention mindset induction on a brief intervention to increase risk perception and reduce alcohol use among university students: A pilot randomized controlled trial

PLOS ONE

Dear Dr. Odenwald,

Thank you for submitting your manuscript to PLOS ONE, and for your careful and thoughtful attention to the comments given by the previous reviewers. Your paper has merit but does not fully meet PLOS ONE’s publication criteria as it currently stands. We invite you to submit a revised version of the manuscript that addresses the points raised during the review process. There has been a bit of a reshuffle of editors, and we have consulted a statistics specialist who makes a few excellent suggestions which would enhance your manuscript. Could I also ask that you carefully check that you have met the CONSORT reporting guidance exactly (using the elaboration statement if needed) and I will check this carefully on resubmission (e.g. as the reviewer states, the abstract should be structured).

We look forward to receiving your revised manuscript.

Kind regards,

Dr Gillian W. Shorter

Academic Editor

PLOS ONE

Additional Editor Comments (if provided):

Could you thoroughly check that you have addressed the full CONSORT statement, checking the elaboration statement that the required detail is present. I will check this carefully on resubmission. Thank you.

Reviewers' comments:

Reviewer's Responses to Questions

**Comments to the Author**

1. If the authors have adequately addressed your comments raised in a previous round of review and you feel that this manuscript is now acceptable for publication, you may indicate that here to bypass the “Comments to the Author” section, enter your conflict of interest statement in the “Confidential to Editor” section, and submit your "Accept" recommendation.

Reviewer #2: All comments have been addressed

Reviewer #3: Minor comments below.

2. Is the manuscript technically sound, and do the data support the conclusions?

Reviewer #2: Yes

Reviewer #3: Yes

3. Has the statistical analysis been performed appropriately and rigorously? 

Reviewer #2: Yes

Reviewer #3: Yes

4. Have the authors made all data underlying the findings in their manuscript fully available?

Reviewer #2: Yes

Reviewer #3: Yes

5. Is the manuscript presented in an intelligible fashion and written in standard English?

Reviewer #2: Yes

Reviewer #3: Yes

6. Review Comments to the Author

Reviewer #2: I am satisfied with the authors' responses to my comments as well as the changes made in the manuscript. The language is improved, but I did notice an error on p. 13, line 463 where the word "of" is missing ("Please note, however, that because how we designed..."; should be "Please note, however, that because of how we designed..."). There may be additional similar minor errors so a final proofreading is probably a good idea. Good luck with your future research!

Reviewer #3: Important note: This review pertains only to ‘statistical aspects’ of the study and so ‘clinical aspects’ [like medical importance, relevance of the study, ‘clinical significance and implication(s)’ of the whole study, etc.] are to be evaluated [should be assessed] separately/independently.

It is definitely a well written article; however, there are few questions and I have the following minor comments to make:

Your ABSTRACT is well drafted but essay type. Please note that it is preferable [refer to item 1b of CONSORT checklist 2010: Structured summary of trial design, methods, results, and conclusions] to divide the ABSTRACT with small sections like ‘Objective(s)’, ‘Methods’, ‘Results’, ‘Conclusions’, etc. which is a accepted practice of most good/standard journals [including PLOS]. It will definitely be more informative then.

Please check {for a possible} typing error in line 120 [The trial started in the winter term 2017/18, recruitment was originally planned for two subsequent semesters between November 2017 and October 2012 and t2 assessments were planned to be terminated before the end of the teaching term.].

To provide a description of baseline characteristics is entirely reasonable (since it is clearly important in assessing to whom the results of the trial can be applied), however, it does not require the division of baseline characteristics by treatment groups {lines 223-4: To test for baseline differences between mindset groups, χ²-tests were performed for categorical variables and univariate ANOVAs for continuous variables}. Statistical comparison of baseline characteristics [last ‘P-value’ column in Table 1] is not desirable at all [because even if P-value turns out to be significant (while comparing baseline characteristics despite random allocation), it is, by definition, a false positive] as you are supposed to be testing ‘randomization’ then, which in any single trial may not balance all baseline characteristics because ‘randomization’ is a sort of ‘insurance’ and not a guarantee scheme.

References:

1. Stuart J. Pocock, et al., ‘Subgroup analysis, covariate adjustment and baseline comparisons in clinical trial reporting: current practice and problems’, Statistics in medicine, 2002; 21:2917–2930 [Particularly page 2927]

2. Harrington D, et al., ‘New guidelines for statistical reporting in the journal’, N Engl J Med 2019;381:285-6

[Important message from these articles: Never do any comparison with respect to ‘baseline’ characteristics].

Because for missing data due to dropout (two participants), the last observation carried forward method was chosen [lines 218-9], please note (must be known to these learned authors, still may please be noted) that according to available literature [example, “Inference and Missing Data,” Biometrika, 1976, vol:63, 581–592 and “Multiple Imputation After 18+ Years,” Journal of the American Statistical Association, 1996, vol:91, 473–489] ‘Multiple Imputation’ technique [procedure replaces each missing value with a set of plausible values that represent the uncertainty about the right value to impute. These multiply imputed data sets are then analysed by using standard procedures for complete data and combining the results from these analyses. It is claimed that ‘No matter which complete-data analysis is used, the process of combining results from different imputed data sets is essentially the same’. This results in valid statistical inferences that properly reflect the uncertainty due to missing values], is preferred [considering MCAR (Missing Completely At Random) expected nature of data] than {despite being time-consuming and involving much more computations} compare to all out of other important imputation techniques frequently used [like Group Means, Hot-deck Imputation, Baseline Observation Carried Forward (BOCF), Worst Observation Carried Forward (WOCF), Predicted Mean, and even Last Observation Carried Forward (LOCF)].

When you say [line 23] that “In contrast to our hypotheses, …….” and later in lines 82-91 (not pasted) & in lines 307-9 [We had expected that the induction of a deliberative versus an implemental mindset would enhance the acceptance of the alcohol feedback and, thus, increase participants’ risk perceptions and readiness to change, leading to reductions in alcohol consumption], remember the principle of ‘equipoise’ [which means that there is genuine uncertainty in the expert medical community over whether a treatment will be beneficial. An ethical dilemma arises in a clinical trial when the investigator(s) begin to believe that the treatment or intervention administered in one arm of the trial is significantly outperforming the other arms. A trial should begin with a null hypothesis, and there should exists no decisive evidence that the intervention or drug being tested will be superior to existing treatments or effective at all. Infact equipoise is a sort of assurance that there is no prejudice and it is to avoid Asher’s paradox.].

While saying [in lines 26-27] “neither deliberative nor implemental mindset participants showed any changes in risk perceptions or in their readiness to change alcohol consumption” remember that “Absence of evidence is not evidence of absence” [Altman DG, Bland JM. BMJ volume 311, 1995, p 485 (Reprinted : Australian Veterinary Journal 1996;74, 311)]. Absence of evidence implies that null hypothesis of no difference is not rejected i.e. result is not significant but that does not amount to evidence of absence i.e. evidence of ‘no difference’.

Justification {or account given otherwise} on sample size given in lines 122-27 was not necessary because this being a pilot study sample size is not a vital issue anyway. As said in lines 169-70 [For SOCRATES and the URICA Precontemplation Scale, we modified the original Likert answer scales into a visual analogue scales to prevent memory biases] can you give more explanation on ‘how modification of the Likert answer scales into a visual analogue scales’ prevent memory biases? (and what these memory biases are)?

Note that non-parametric parallel of one-way ANOVA is ‘Kruskal-Wallis’ test and not Mann-Whitney test [as said in lines 224-5: If Levene tests revealed heterogeneity of variances, Mann-Whitney tests were used instead.]. Mann-Whitney test is for comparing two groups and parallel to ‘t’ test for two independent groups. In this trial there are two independent groups and application of Mann-Whitney test [when Levene test revealed heterogeneity of variances] is correct. But when there are two independent groups to be compared ‘why apply ANOVA’ which is for comparison of more than two groups? {it is true that ‘if there are only two independent groups ‘ANOVA’s “F” is square of ‘t’ but logic and so algorithm are different}. Whenever there are two independent groups we never apply ANOVA, though ‘F’ and ‘t’ are mathematically similar. Please explain if the reason is what said in lines 231-2 [“We chose ANOVAs as many findings speak for the robustness of the analysis of variance concerning violated assumptions”], {what is that?}.

I wish that a larger study over coming limitations of the present study [as pointed out in lines 400-410] should be conducted by the same team of scientists.

7. PLOS authors have the option to publish the peer review history of their article (what does this mean?). If published, this will include your full peer review and any attached files.

Reviewer #2: Yes: Anne H Berman

Reviewer #3: Yes: Dr. Sanjeev Sarmukaddam

---

## [Author Response · Author response to Decision Letter 1]

18 Aug 2020

Please find the detailed responses to the reviewers' comments in an uploaded file.

---

## [Decision Letter · Decision Letter 2]

26 Aug 2020

The effects of pre-intervention mindset induction on a brief intervention to increase risk perception and reduce alcohol use among university students: A pilot randomized controlled trial

PONE-D-19-18841R2

Dear Dr. Odenwald,

We’re pleased to inform you that your manuscript has been judged scientifically suitable for publication and will be formally accepted for publication once it meets all outstanding technical requirements.

Kind regards,

Dr Gillian W Shorter

Academic Editor

PLOS ONE

Additional Editor Comments (optional):

Reviewers' comments:

Reviewer's Responses to Questions

**Comments to the Author**

1. If the authors have adequately addressed your comments raised in a previous round of review and you feel that this manuscript is now acceptable for publication, you may indicate that here to bypass the “Comments to the Author” section, enter your conflict of interest statement in the “Confidential to Editor” section, and submit your "Accept" recommendation.

Reviewer #3: All comments have been addressed

2. Is the manuscript technically sound, and do the data support the conclusions?

Reviewer #3: Yes

3. Has the statistical analysis been performed appropriately and rigorously? 

Reviewer #3: Yes

4. Have the authors made all data underlying the findings in their manuscript fully available?

Reviewer #3: (No Response)

5. Is the manuscript presented in an intelligible fashion and written in standard English?

Reviewer #3: Yes

6. Review Comments to the Author

Reviewer #3: COMMENTS: I am happy to learn that the numeric results changed very little [due to analysis using multiple imputation instead of the Last Observation Carried Forward Method & so the outcome of statistical tests stayed the same with no further implications for the interpretation], however, it is always good to use correct/right/desirable methods. Response about issues raised/made on earlier draft by me regarding the baseline characteristics and the ANOVA are positively addressed. I am satisfied and, in my opinion, the manuscript is improved a lot. I recommend acceptance, without any hesitation, as now it has achieved acceptable level of our journal, in my opinion.

7. PLOS authors have the option to publish the peer review history of their article (what does this mean?). If published, this will include your full peer review and any attached files.

Reviewer #3: **Yes: **Dr. Sanjeev Sarmukaddam

---

## [Editor Report · Acceptance letter]

8 Sep 2020

PONE-D-19-18841R2 

The effects of pre-intervention mindset induction on a brief intervention to increase risk perception and reduce alcohol use among university students: A pilot randomized controlled trial 

Dear Dr. Odenwald:

I'm pleased to inform you that your manuscript has been deemed suitable for publication in PLOS ONE. Congratulations! Your manuscript is now with our production department. 

Kind regards, 

on behalf of

Dr. Gillian Shorter 

Academic Editor

PLOS ONE